# Influence of Age on Partial Clinical Remission among Children with Newly Diagnosed Type 1 Diabetes

**DOI:** 10.3390/ijerph17134801

**Published:** 2020-07-03

**Authors:** Stefano Passanisi, Giuseppina Salzano, Albino Gasbarro, Valentina Urzì Brancati, Matilde Mondio, Giovanni Battista Pajno, Angela Alibrandi, Fortunato Lombardo

**Affiliations:** 1Department of Human Pathology in Adult and Developmental Age “Gaetano Barresi”, University of Messina, Via Consolare Valeria 1, 98124 Messina, Italy; spassanisi87@gmail.com (S.P.); gsalzano@unime.it (G.S.); docalbigsb@gmail.com (A.G.); valoussene@hotmail.it (V.U.B.); matmon62@tiscali.it (M.M.); gpajno@unime.it (G.B.P.); 2Department of Economics, Unit of Statistical and Mathematical Sciences, University of Messina, 98124 Messina, Italy; aalibrandi@unime.it

**Keywords:** C-peptide, early diagnosis, length, GADA, honeymoon

## Abstract

Partial clinical remission (PCR) is a transitory period characterized by the residual endogenous insulin secretion following type 1 diabetes (T1D) diagnosis and introducing the insulin therapy. Scientific interest in PCR has been recently increasing, as this phase could be crucial to preserve functional beta cells after T1D onset, also taking advantage of new therapeutic opportunities. The aim of this study was to assess the frequency, duration and associated factors of PCR in children newly diagnosed with T1D. Our cohort study included 167 pediatric patients aged 13.8 ± 4.1 years. The association of clinical and laboratory factors with the occurrence and duration of PCR was evaluated via logistic regression and multivariable generalized linear model, respectively. PCR occurred in 63.5% of the examined patients. Patients who achieved the remission phase were significantly older, and they had lower daily insulin requirement compared with non-remitters. PCR was positively associated to body mass index (OR = 1.11; *p* = 0.032), pH value (OR 49.02; *p* = 0.003) and c-peptide levels (OR 12.8; *p* = 0.002). The average duration of PCR was 13.4 months, and older age at diagnosis was the only predictor factor. Two years after diagnosis remitter patients had lower HbA1c and daily insulin requirement.

## 1. Introduction

Type 1 diabetes (T1D) is a chronic metabolic disease characterized by a selective and progressive decline of pancreatic β cells functions that leads to an absolute insulin deficiency [1]. The diagnosis of T1D is often followed by a partial clinical remission (PCR), also known as “honeymoon phase”. This period is characterized by the reduction of the glucose toxicity because of the introduction of subcutaneous insulin therapy, which may improve the residual endogenous insulin secretion and decelerate the process of β-cells destruction. Consequently, the patient’s need for exogenous insulin decreases while a good metabolic control is maintaining [2]. PCR may begin within few days or weeks after the introduction of insulin therapy, and it can last from a few weeks to several months [3].

In recent decades, the definition of PCR has hugely varied [4]. One of the first definitions was given by Agner et al. [5] who declared partial remission as that clinical condition characterized by an insulin requirement of ≤50% of the standard daily insulin dose at the discharge from the hospital after the diagnosis of T1D. Most authors agreed with the definition of PCR as insulin requirement of <0.5 IU/kg/day [6,7]. Other authors defined it as the combination of low insulin requirements (0.3–0.5 IU/kg/day) and optimal metabolic control assessed by the value of glycated hemoglobin (HbA1c) value (<7.5% or <4 SDS of normal range) [8,9]. The International Society for Pediatric and Adolescent Diabetes (ISPAD) guidelines 2006/2007 defined the honeymoon phase as the period when daily insulin dosage was less than 0.5 IU per kilogram and HbA1c was less than 7% (53 mmol/mol) [10]. In 2008, the Hvidovre Study Group introduced a novel definition for honeymoon phase: PCR was identified by the phase when the value of insulin dose-adjusted HbA1c (IDAA1c) was ≤9. IDAA1c was calculated by the following formula: HbA1c (%) + (4 × insulin dose (units per kilogram per 24 h)) [11]. Successively, the Danish Study Group [12] validated this measure. The definition of complete remission, known as restoration of normal fasting and postprandial blood glucose without any insulin administration for more than 2 weeks or as insulin requirements <0.2 IU/kg/day combined with HbA1c less than 6% [13,14], is no longer accepted.

Results from the Diabetes Control and Complications Trial had already demonstrated that patients who entered PCR, known as remitters, had a better long-term glycemic control and a lower risk of severe hypoglycemia [15,16]. Several reports confirmed that PCR is associated with reduced prevalence of chronic microvascular and macrovascular complications of T1D [17,18]. Scientific interest in PCR has been recently increasing since it is thought that this phase could be crucial to preserve functional beta cells after T1D onset, also taking advantage of new therapeutic opportunities [19].

Several factors have been found to influence the frequency and the duration of the PCR period including demographic (age, gender, ethnicity, socioeconomic status), clinical (degree of metabolic decompensation, c-peptide levels, presence of autoantibodies, HbA1c levels at the time of diagnosis) and biochemical (nicotinamide, interferon gamma, interleukin10, interleukin 1 receptor type 1) factors [20,21,22]. However, the influence of all these factors remains to be clarified. 

### Aim of the Study

The aim of the present observational study was to assess the frequency and duration of PCR in a cohort study population of children and adolescents newly diagnosed with T1D over a 24-month-follow-up period after diagnosis. Our investigation also aimed to estimate the association of selected clinical and laboratory factors with the occurrence and duration of PCR.

## 2. Material and Methods

The study group included a cohort of children and adolescents from Southern Italy diagnosed with T1D in the years 2012 to 2017 and followed up in our regional Pediatric Diabetes Center in Messina for at least two years. Written informed consent from the parents of each child and adolescent involved in the research study was obtained before the start of study procedures. The study was conducted in accordance with the Helsinki Declaration. All the patients were enrolled in the study at the time of diagnosis. Out of 219 patients who were diagnosed in our Pediatric Diabetes Clinic, 167 (76.3%) completed the study. The remaining patients were excluded since they were followed-up in other Diabetes Centers within two years after the diagnosis.

The diagnosis of T1D was made according to the current criteria of the International Society for Pediatric and Adolescent Diabetes Clinical Practice Consensus Guidelines: classic symptoms of diabetes or hyperglycemic crisis, with plasma glucose concentration ≥11.1 mmol/L (200 mg/dL) or fasting plasma glucose ≥7.0 mmol/L (≥126 mg/dL) or two hour postload glucose ≥11.1 mmol/L (≥200 mg/dL) during an oral glucose toleration test or HbA1c > 6.5% (48 mmol/mol) [23]. In addition, all selected patients met at least one of the following criteria: diabetic ketoacidosis (DKA) at the onset of diabetes, detection of one or more T1D-associated antibodies (glutamic acid decarboxylase, protein tyrosine phosphatase, islet cell, insulin, anti-β-cell-specific zinc transporter 8 autoantibodies) or on-going requirement for subcutaneous insulin therapy.

According to the Italian Society of Pediatric Endocrinology and Diabetes recommendations, all young patients newly diagnosed with T1D are hospitalized for approximately 7 to 10 days in a regional Pediatric Diabetes Center. During this time, after the resolution of acidosis if present, daily insulin dosage is titrated in order to obtain a good metabolic control. Multiple daily insulin injection according to the “basal-bolus” scheme is the chosen regimen in our Center to start insulin treatment. The hospitalization is also necessary for patients without metabolic decompensation at T1D onset to ensure that their parents receive an appropriate diabetes education program.

Data were collected at diagnosis (sex, age, anthropometry (stature, weight, body mass index), presence of DKA, HbA1c, basal c-peptide levels, stimulated c-peptide levels after glucagon stimulation test, T1D associated-antibodies titration, human leucocyte antigens (HLA) typing, total daily insulin dose) and then at 3, 6, 9, 12, 15, 18 and 24 months after diagnosis (anthropometry, HbA1c and total daily insulin dose). Diabetic ketoacidosis (DKA) at diagnosis was identified as blood glucose > 11 mmol/L (200 mg/dL), venous pH < 7.3 or bicarbonate < 15 mmol/L, presence of ketonemia and ketonuria. The severity of DKA was categorized by the degree of acidosis: “mild” if venous pH < 7.3 or bicarbonate < 15 mmol/L, “moderate” if pH < 7.2 and bicarbonate < 10 mmol/L, “severe” if pH < 7.1 and bicarbonate < 5 mmol/L [24]. HbA1c was measured on a DCA Vantage Analyzer (Siemens^®^, Deerfield, Illinois, USA) using high-performance liquid chromatography technique.

C-peptide levels determination is known to be a low-cost and easy-to-use method to assess pancreatic beta-cell function. The glucagon stimulation test was performed after the acidosis had resolved and subcutaneous insulin therapy had been initiated. Serum c-peptide levels were measured before and 6 min after intravenous injection of 0.1 mg pro Kilo (maximum 1 mg) of glucagon (basal and stimulated C-peptide, respectively). Samples were preserved below 0 °C and analyzed by radioimmunoassay (ECLIA) method with the C6000 instrument (Creative-Biolabs, Hamburg, Germany). The c-peptide levels were defined in terms of ng/mL. The normal range for basal c-peptide levels is 0.51–2.72 ng/mL. After the glucagon stimulation test, the c-peptide values can increase up to three fold.

Anthropometric parameters as well as total daily insulin dose were established using data at the time of the hospital discharge. Standard deviation score (SDS) for anthropometric variables was considered. In the present paper, we only reported information on body mass index (BMI) since it is the most investigated anthropometric factor related to the occurrence of remission phase. Total daily insulin requirements of patients were expressed in IU/Kg. Regarding T1D-associated antibodies, we only considered glutamic acid decarboxylase antibodies (GADA) and islet cell antibodies (ICA) because of incomplete other antibodies results. At the time of our study procedures, protein tyrosine phosphatase and insulin autoantibodies were not routinely used in the diagnostic work-up in patients newly diagnosed with diabetes, and anti-β-cell-specific zinc transporter 8 antibodies as markers of diabetes autoimmunity were identified after the start of study population recruitment. The presence of ICA was determined by an immunofluorescence assay, and GADA were measured with enzyme linked immunosorbent assay.

Finally, HLA was considered predisposing to T1D in the event of presence of specific DR/DQ alleles (e.g., DRB1*03-DQB1*0201 (DR3) or DRB1*04-DQB1*0302 (DR4)).

### 2.1. Definition of Partial Clinical Remission 

PCR was determined according to the IDAA1c definition: HbA1c + (4 × insulin dose), where HbA1C is expressed in % and insulin dose in units/kg/day. The presence of PCR was defined as a value ≤9 [11]. Among all the above definitions for PCR, this measure represents the most validated and scientifically accepted. PCR was evaluated starting from the three-month follow-up visit after diagnosis and then every 3 months.

### 2.2. Statistical Analysis 

Numerical variables were expressed as mean and standard deviation and the categorical variables as absolute frequency and percentage. The non-parametric approach was used for statistical analysis, since most numerical variables were not normally distributed, such as verified by Kolmogorov–Smirnov test. The comparison between remitters and non-remitters was performed using Mann–Whitney test for numerical variables and chi-square test for categorical variables.

For all participants, univariate and multivariate binary logistic regression models were estimated in order to assess the dependence of remission (Yes or No) on potentially explicative variables such as age at the onset, gender, BMI, presence of DKA, pH, HbA1c, c-peptide levels, HLA predisposition, ICA and GADA positivity, total daily insulin dose.

Focusing the attention on the remitter patients, we evaluated the PCR duration among three age classes (0–4 years, 5–9 years and 10–16 years). The Jonckheere–Terpstra test was applied in order to assess if the PCR duration significantly increased with age increasing. 

The PCR duration was compared among remitters according to the presence of DKA at the onset, HLA predisposition and T1D-associated antibodies using the same Mann–Whitney test. Finally, we estimated a multivariable generalized linear model with robust standard error to identify significant predictors of PCR duration, taking into account all examined covariates, adjusting for sex and age, at T1D diagnosis of patients. Statistical analyses were performed using SPSS 22.0 (IBM, New York, NY, USA) for Windows package. A *p*-value smaller than 0.050 was considered to be statistically significant. 

## 3. Results

Our study population consisted of a cohort of 167 pediatric patients with a slight prevalence of male gender (52.7%). T1D was diagnosed at the mean age of 8.6 ± 3.8 years. Over half of the patients (51%) experienced DKA at the onset. Mean HbA1c value at the diagnosis of T1D was 10.7 ± 1.9% (93 ± 21 mmol/mol). As expected, basal and stimulated c-peptide mean levels were below the normal value ranges, respectively, 0.43 ± 0.38 and 0.77 ± 0.64 ng/mL. Almost all the patients (87.4%) had a genetic predisposition to T1D as demonstrated by the presence of HLA-DR3 and/or DR4. Most patients (65.6%) showed GADA positivity, whereas ICA were present in 50.7% of the recruited patients. Mean total daily insulin dose, evaluated at the discharge (approximately after 5–7 days from the introduction of subcutaneous insulin treatment) was 0.75 ± 0.31 IU/Kg. Demographic and clinical characteristics of study participants are summarized in Table 1.

### 3.1. Partial Clinical Remission Occurrence and Associated Variables at T1D Diagnosis

PCR occurred in 106 (63.5%) of patients, and there was no difference in gender distribution (58 boys and 48 girls). There also was no significant difference in BMI SDS and HbA1c value at the T1D onset. Patients who entered the PCR were significantly older than the other group (9.3 ± 3.7 vs. 7.4 ± 3.7 years, *p* = 0.002). Age at T1D onset also differed between two groups when subgroups based on age classes (0–4 years, 5–9 years, 10–16 years) were compared (*p* = 0.011). DKA prevalence was significantly lower in patients with PCR than patients without PCR (respectively, 43.2% and 58.9%, *p* = 0.044). Furthermore, among patients with DKA at the onset, PCR was more frequent in those individuals with milder DKA than in the subjects with moderate or severe DKA at onset (*p* = 0.015). Patients who experienced the PCR had significantly higher pH at T1D diagnosis than patients without PCR (7.3 ± 0.12 vs. 7.23 ± 0.16, *p* = 0.005). Other parameters that differed between the groups were both basal and stimulated c-peptide levels (0.51 ± 0.42 vs. 0.29 ± 0.22 ng/mL, *p* < 0.001 and 1.01 ± 0.73 vs. 0.49 ± 0.35 ng/mL, *p* < 0.001, respectively). There were no significant differences in HLA predisposition between the groups, as well as in the positivity of GADA and/or ICA (Table 2).

The occurrence of PCR was associated with age at diagnosis (*p* = 0.002; OR 1.14; CI 1.05–1.25), BMI SDS (*p* = 0.032; OR = 1.11; CI 1.01–1.22), blood pH value (*p* = 0.003; OR 49.02; CI 3.63–662.1), c-peptide levels (*p* = 0.002; OR 12.8; CI 2.54–64.47) and total daily insulin requirement (*p* = 0.028; OR 0.27; CI 0.08–0.87). Multivariate analysis of parameters assessed at T1D onset revealed that the presence of GADA (*p* = 0.022; OR 24.65; CI 1.6–380.66) and total daily insulin dose (*p* = 0.031; OR 0.06; CI 0.01–0.77) were independent predictors of PCR occurrence (Table 3).

### 3.2. Partial Clinical Remission Duration

The duration of PCR was 13.4 ± 6.9 (3–24) months. The most important factor that influenced PCR duration was the age at T1D diagnosis. The Jonckheere–Terpstra test revealed that the duration of PCR was longer in accordance with age increasing (*p* = 0.045) (Figure 1). Multivariable GLM showed that only age at T1D onset (*p* = 0.005; B 0.84; CI 0.25–1.43) was an independent predictor of PCR duration (Table 4). The occurrence of DKA at T1D onset, as well as HLA predisposition and autoantibodies positivity, did not influence the duration of PCR.

### 3.3. Anthropometric Parameters, HbA1c and Total Daily Insulin Dose at 1 and 2 Years after T1D Onset

Daily insulin requirement and HbA1c significantly differed between the groups over the 24-month follow-up. HbA1c, evaluated as the average value of the last 12 months, was significantly lower in patients with PCR both after 1 year (6.6 ± 0.6 vs. 7.2 ± 0.6, *p* < 0.001) and after 2 years from diagnosis (7.0 ± 0.7 vs. 7.4 ± 0.8, *p* = 0.017). The total daily insulin dose also was lower in patients who entered PCR (0.45 ± 0.25 vs. 0.78 ± 0.3, *p* < 0.001 at 1 year-follow-up visit; 0.63 ± 0.27 vs. 0.87 ± 0.23, *p* < 0.001 at 2-year-follow-up visit). No significant differences were detected in BMI SDS.

## 4. Discussion

Our study revealed that 63.5% of our pediatric cohort study population entered the PCR. According to the data reported in the literature, the partial remission rate seems to vary considerably between countries. A longitudinal observational study from 255 centers in Germany and Austria demonstrated that partial remission occurred in 71% of a large study cohort patients (3657 children and adolescents) with new-onset T1D who were continuously followed over 6 years [25].

Two recent studies conducted in the U.S. showed lower the remissions rates. Particularly, Marino et al. reported 42.8% of remitters among 204 young patients (2–14 years) who were retrospectively analyzed, whereas 35.8% of remitters were described in another longitudinal retrospective study including 123 subjects with T1D of 4–5 years of duration [26,27,28]. In Poland, PCR prevalence was estimated in 61.8% of 186 patients newly diagnosed with T1D and followed-up over 24 months and in 59% of 194 children with at least 4 years of T1D duration [2,29]. Chiavaroli et al. reported an overall rate of partial remission at 3 months of 42.4% in a cohort study of 678 New Zealand patients aged <15 years [30]. A report from Sweden described a rate of 80% of remitters among 149 children and adolescents (0–16 years) with new-onset of T1D [9].

Our study showed that PCR was positively influenced by blood pH, c-peptide levels and BMI SDS. Regarding the results on blood pH and c-peptide levels, they could be closely related. Blood pH levels are generally linked to the patient’s clinical condition at the time of T1D diagnosis, whereas c-peptide levels reflect the functional capacity of the remaining beta cells. Early recognition of T1D symptoms leads to precocious diagnosis in children who presented with quite good clinical state in the absence of acidosis, as demonstrated by the normality of pH value. Furthermore, in the event of early diagnosis, a large amount of residual beta cells is preserved, and therefore, higher c-peptide levels are present. In addition, insulin treatment that is started directly at onset sustains the function of the remaining beta cells, and thus, the occurrence of PRC becomes more likely [31].

Our finding that BMI SDS was positively associated to PCR is in agreement with other previous studies [2,21]. This finding could be explained by the “acceleration hypothesis”. This theory supposes that higher BMI, which is suggestive of insulin resistance, causes a worsening in glycemic control. Therefore, higher BMI accelerates the appearance of clinical condition leading to an early diagnosis and, consequently, to the immediate introduction of insulin therapy. Insulin treatment decreases insulin resistance, and the production of the residual beta-cells starts becoming sufficient again, even though the destruction process continues [32].

The prevalence of PCR was significantly lower in children diagnosed while younger than 5 years. This finding is consistent with the results of numerous other studies [3,13,21,33]. It is known that the preschool children had a high frequency of DKA related to shorter duration of symptoms, which are usually poorly recognized, prior to diagnosis. In younger children, the process of beta cell destruction is faster compared to older age group, causing a more severe metabolic decompensation with less residual beta cell function [34,35]. Furthermore, Atkinson et al. revealed that younger age at T1D onset was associated with higher levels of pro-inflammatory cytokines (e.g., CD20^+^B cells, CD45^+^cells, CD8^+^T cells) leading to beta-cell destruction [36].

Our study showed that total daily insulin requirement at hospital discharge (approximately 7–10 days after the diagnosis) was an independent predictor of PCR occurrence. Therefore, the process leading to the achievement of PCR seems to be so precocious as to allow a prompt reduction of the daily insulin dose from the first days of the disease. Similarly to our study, Kara et al. found that remitter patients also had lower basal insulin requirements, suggesting that patients who entered the PCR could easily meet their basal insulin requirements due to residual endogenous insulin secretion in the early phase of the disease [37].

Our finding that the presence of GADA was another independent predictor of PCR could be discussed. The presence of markers of islet cell autoimmunity is usually considered a factor influencing the decline of beta-cell function [38]. However, a recent report on PCR in Brazilian children and adolescents revealed that the presence of DRB1*03-DQB1*0201 alleles was more prevalent among patients entering remission. Interestingly, all patients carrying this HLA genotype had positivity for GADA at diagnosis, most of them as a single autoantibody [39]. This association between haplotype of class II human leukocyte antigen and synthesis of autoantibodies could partially explain our results, even though further studies to better investigate this topic are awaited.

The mean duration of PCR in our study population was 13.4 months. Our data are controversial with those of other researchers who reported an average duration of this phase of 7 months [31]. The longest mean duration (26.9 months) was previously described in Turkey [40]. Recently, a retrospective study showed that patients with a longer duration of PCR seem at risk of developing another autoimmune disease during a 10-year follow-up period after diagnosis [41]. Our results revealed that age at diagnosis was the only predictor of PCR duration. Furthermore, the duration of PCR seems to increase with age increasing. Several studies had already observed that age at diagnosis was a factor in strong association with the remission rate, as discussed above. Instead, only Abdoul-Rasoul et al. [21] have so far described the association between age at diagnosis and the duration of remission phase. This finding could be explained by the following underlying mechanisms. Older patients have a slower autoimmune derangement process and more functional beta-cells mass compared with younger children. Moreover, adolescents may have a more remarkable insulin sensitivity, especially after puberty.

We observed that PCR might have a beneficial effect on the course of T1D. Patients who achieved the PCR had lower HbA1c levels over the 24-monthfollow-up period. Similar conclusions were also reported by other authors [21,29]. These findings further confirm the need to identify with certainty clinical and laboratory biomarkers, which are associated with PCR in order to ensure a better metabolic control for patients with T1D from the first years of disease.

A limitation of our study was the absence of evaluation of all T1D-associated antibodies. Unfortunately, the start of our studies procedures was prior to the introduction of zinc transporter 8 antibodies in the diagnostic work-up of autoimmune diabetes, as well as data on autoantibodies to protein tyrosine phosphatase and insulin were incomplete. Moreover, we did not consider pubertal stage at diagnosis, which could have provided additional information to investigate the correlation between age at T1D onset and PCR duration. It is known that puberty is characterized by physiological decreased insulin sensitivity. However, previous studies reported conflicting results on the real influence of pubertal stage in determining the occurrence of PCR [37,42].

## 5. Conclusions

Our study shows that almost two thirds of pediatric patients newly diagnosed with T1D experience partial remission phase. Clinical and laboratory data, such as BMI SDS, blood pH, c-peptide levels and older rage at diagnosis facilitate the achievement of PCR.

The mean duration of PCR is more than 12 months, and according to the present study, age at diagnosis is the only parameter that may influence the length of PCR. Particularly, the PCR duration significantly increases with age at diagnosis increasing.

Patients who enter the PCR have a better metabolic compensation in the first years of life compared with the other patients, as demonstrated by lower HbA1c levels and total daily insulin requirements after two-year follow-up. Better metabolic control is known to be associated with reduced risk of developing long-term diabetic complications. Therefore, our results confirm the potential protective role of PCR for microvascular and macrovascular complications of T1D.

The remission process is a crucial phase in the clinical course of T1D, and its scientific interest remains unchanged over the years. However, some aspects need to be clarified and further studies, with larger pediatric cohort study populations and long-lasting follow-up, are awaited in order to better understand the metabolic and immunological mechanisms involved in the epidemiology, diagnostic criteria, duration, natural history and clinical outcomes of partial remission phase.

## Figures and Tables

**Figure 1 ijerph-17-04801-f001:**
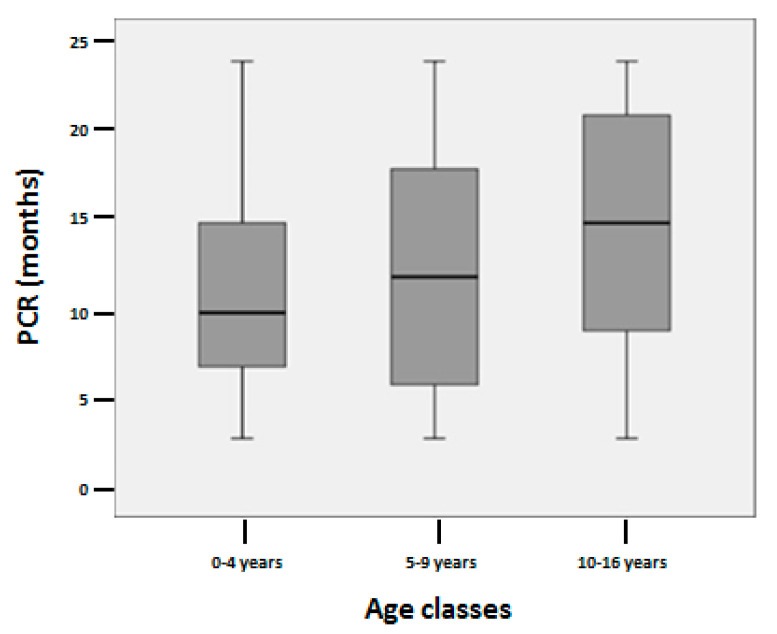
Boxplot illustrating distribution of PCR duration among age groups.

**Table 1 ijerph-17-04801-t001:** Descriptive statistics for categorical (percentages) and numerical (mean ± SDS) variables of our study population.

Variables	Percentages and Mean ± SDS	Median (IQR)
Gender		
Male	52.70%	
Female	47.30%	
Age at the onset (years)	8.6 ± 3.8	8.8 (5.9; 11.5)
Age classes at the onset		
0–4 years	21%	
5–9 years	41.30%	
10–16 years	37.70%	
BMI at the onset (SDS)	−0.08 ± 1.21	−0.04 (−0.91; 0.79)
pH	7.28 ± 0.14	7.31 (7.22; 7.37)
Presence of DKA		
Yes	49%	
No	51%	
HbA1c at the onset (%)	10.7 ± 1.9	10.4 (9.5; 12.1)
HbA1c at the onset (mmol/mol)	93 ± 21	90 (80; 109)
Basal c-peptide (ng/mL)	0.43 ± 0.38	0.26 (0.10; 0.50)
Stimulated c-peptide (ng/mL)	0.77 ± 0.64	0.58 (0.39; 0.97)
HLA predisposition		
Yes	87.40%	
No	12.60%	
GADA positivity		
Yes	65.60%	
No	34.40%	
ICA positivity		
Yes	50.70%	
No	49.30%	
Total daily insulin dose at the onset (IU/Kg)	0.75 ± 0.31	0.73 (0.52; 0.96)
PCR occurrence		
Yes	63.50%	
No	36.50%	
BMI at 1-year-follow-up (SDS)	0.15 ± 0.98	0.9 (−0.5; 0.85)
Last year mean value HbA1c at 1-year-follow-up (%)	6.8 ± 0.7	6.8 (6.4; 7.3)
Last year mean value HbA1c at 1-year-follow-up (mmol/mol)	51 ± 6.9	51 (46; 56)
Total daily insulin dose at 1-year-follow-up (IU/Kg)	0.57 ± 0.31	0.45 (0.40; 0.65)
BMI at 2-years-follow-up (SDS)	0.16 ± 0.98	0.1 (−0.45; 0.85)
Last year mean value HbA1c at 2-years-follow-up (%)	7.2 ± 0.8	7.1 (6.7; 7.5)
Last year mean value HbA1c at 2-years-follow-up (mmol/mol)	55 ± 8.4	54 (50; 58)
Total daily insulin dose at 2-years-follow-up (IU/Kg)	0.71 ± 0.28	0.65 (0.51; 0.78)

**Table 2 ijerph-17-04801-t002:** Demographic and clinical variables of patients who entered partial remission or not.

	Remitters		Non-Remitters		
Variables	Frequency or Mean ± SDS	Median (IQR)	Frequency or Mean ± SDS	Median (IQR)	*p-*Value
N	63.5%		36.5%		
Sex					0.49
Male	54.70%		49.20%		
Female	45.30%		50.80%		
Age at the onset (years)	9.3 ± 3.7	9.4 (6.8; 12.3)	7.4 ± 3.7	7.9 (3.9; 10.2)	0.002
Age classes					0.011
0–4 years	14.20%		32.80%		
5–9 years	42.50%		39.30%		
10–16 years	43.40%		27.90%		
BMI at the onset (SDS)	−0.04 ± 1.10	0.01 (−0.81; 1.06)	−0.31 ± 1.3	−0.08 (−1.23; 0.54)	0.145
Basal c-peptide (ng/mL) at the onset	0.51 ± 0.42	0.30 (0.12; 0.51)	0.29 ± 0.22	0.20 (0.10; 0.37)	<0.001
Stimulated c-peptide (ng/mL) at the onset	1.01 ± 0.73	0.70 (0.55; 1.70)	0.49 ± 0.35	0.40 (0.28; 0.58)	<0.001
pH	7.3 ± 0.12	7.32 (7.27; 7.38)	7.23 ± 0.16	7.28 (7.15; 7.36)	0.005
DKA at the onset					0.044
Yes	43.20%		58.90%		
No	56.80%		41.10%		
Severity of DKA					0.015
Mild	70%		35.50%		
Moderate	15%		32.30%		
Severe	15%		32.30%		
HbA1c at the onset (%)	10.7 ± 1.9	10.4 (9.5; 12.3)	10.6 ± 2.0	10.5 (9.3; 12.1)	0.972
HbA1c at the onset (mmol/mol)	93 ± 21	90 (80; 111)	92 ± 23	91 (78; 109)	0.972
HLA predisposition					0.526
Yes	84.70%		91.70%		
No	15.30%		8.30%		
ICA positivity					0.116
Yes	45.70%		60%		
No	54.30%		40%		
GADA positivity					0.232
Yes	61.40%		73.80%		
No	38.60%		26.20%		
Total daily insulin dose at the onset (IU/Kg)	0.71 ± 0.27	0.72 (0.50; 0.93)	0.83 ± 0.35	0.77 (0.57; 1.16)	0.103
BMI at 1-year-follow-up (SDS)	0.13 ± 1	0.1 (−0.59; 0.92)	0.19 ± 0.95	0.23 (−0.35; 0.79)	0.651
Last year mean value HbA1c at 1-year-follow-up (%)	6.6 ± 0.6	6.7 (6.2; 6.9)	7.2 ± 0.6	7.3 (6.7; 7.7)	<0.001
Last year mean value HbA1c at 1-year-follow-up (mmol/mol)	49 ± 6.5	50 (44; 52)	55 ± 6.1	56 (50; 61)	<0.001
Total daily insulin dose at 1-year-follow-up (IU/Kg)	0.45 ± 0.25	0.35 (0.25; 0.50)	0.78 ± 0.3	0.70 (0.50; 0.90)	<0.001
BMI at 2-years-follow-up (SDS)	0.18 ± 0.96	0.01 (−0.45; 0.86)	0.13 ± 1.01	0.2 (−0.43; 0.74)	0.899
Last year mean value HbA1c at 2-year-follow-up (%)	7.0 ± 0.7	7.0 (6.5; 7.5)	7.4 ± 0.8	7.3 (6.9; 7.7)	0.017
Last year mean value HbA1c at 2-year-follow-up (mmol/mol)	53 ± 8.1	53 (48; 58)	57 ± 8.6	56 (52; 61)	0.017
Total daily insulin dose at 2-years-follow-up (IU/Kg)	0.63 ± 0.27	0.48 (0.35; 0.69)	0.87 ± 0.23	0.80 (0.60; 0.90)	<0.001

**Table 3 ijerph-17-04801-t003:** Results of Univariate and Multivariate Logistic regression models for remission (Yes or No).

	Univariate			Multivariate		
Variables	Crude OR	95% C.I.	*p*-Value	Adjusted OR	95% C.I.	*p*-Value
Age at the onset	1.14	1.05–1.25	0.002	1.15	0.96–1.39	0.139
Gender (male)	1.25	0.67–2.35	0.490	1.85	0.48–7.17	0.377
BMI	1.11	1.01–1.22	0.032	1.1	0.85–1.43	0.478
pH	49.02	3.63–662.1	0.003	0.51	0.01–328.35	0.838
Presence of DKA	0.53	0.27–1.03	0.062	0.8	0.13–4.88	0.812
HbA1c	1.03	0.86–1.22	0.775	0.9	0.64–1.28	0.563
c-peptide levels	12.8	2.54–64.47	0.002	6.72	0.12–372.3	0.352
HLA predisposition	0.51	0.13–2.01	0.331	0.19	0.01–2.69	0.218
ICA positivity	0.56	0.28-1.13	0.104	0.24	0.02–3.01	0.266
GADA positivity	0.57	0.25–1.28	0.566	24.65	1.6–380.66	0.022
Total daily insulin dose	0.27	0.08–0.87	0.028	0.06	0.01–0.77	0.031

**Table 4 ijerph-17-04801-t004:** Results of Generalized Linear Model for HM duration.

Variables	B	95% C.I.	*p*-Value
Age at the onset	0.84	0.25;1.43	0.005
Gender	1.21	−2.69; 5.11	0.543
BMI	0.29	−1.73; 2.32	0.776
pH	5.04	−17.05; 27.12	0.655
Presence of DKA	−0.40	−5.87; 5.07	0.885
C-peptide	−3.06	−8.35; 2.23	0.257
HLA predisposition	−3.74	−9.46; 1.98	0.200
ICA positivity	−0.516	−6.85; 5.82	0.873
GADA positivity	4.15	−2.73; 11.04	0.237
Total daily insulin dose	−5.38	−12.94; 2.17	0.162

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
