# Peer review of "Influence of Age on Partial Clinical Remission among Children with Newly Diagnosed Type 1 Diabetes"

_ijerph, 2020, doi:10.3390/ijerph17134801_

Round 1

Reviewer 1 Report

I read with interest the manuscript # ijerph-841728 entitled “Age at diagnosis remains the main factor influencing the frequency and duration of partial remission in children newly diagnosed with type 1 diabetes: an observational study”

The aim of the present study was to define the prevalence and duration of partial remission in a cohort of paediatric patients suffering from type 1 diabetes. Secondary outcomes were to establish the influence of clinical and/or laboratory parameters on the occurrence and  length of the remission phase.

The authors showed that  63.5% of patients experienced partial remission. The mean duration of the remission phase in this study population was 13.4 months. Age was the main factor influencing both the occurrence and the duration of partial remission. These findings are very interesting and important to collect more information about this crucial phase in the clinical history of type 1 diabetes. I have only minor comments and points for discussion:

  1. The authors cited the criterion for defining the severity of diabetic ketoacidosis. However, data on HCO3 levels at diagnosis of type 1 diabetes were not reported and considered in study procedures.
  2. The authors mentioned the glucagon stimulation test for determining stimulated c-peptide levels. However, other methods for assessing the stimulated c-peptide levels exist. In addition, both baseline and stimulated C-peptide normal values should be added.
  3. Did the authors assess c-peptide levels during the follow-up period?
  4. Several studies concerning partial remission have been published in the literature, but researchers often disagreed with the definition of partial remission. The different approaches used by authors could lead to different conclusions. Have the authors considered this aspect in the discussion of the manuscript?
  5. Recently, it has been shown that the length of partial clinical remission phase is related to the association of autoimmune diseases with type 1 diabetes (doi: 10.1155/2020/2630827). This aspect should be discussed.
  6. Please be consistent with references format (verify Journal rules for writing or not day and month of publication?; last page format : -0; -00; -000, ...?)

Author Response

Reviewer #1

Comments to the Author

I read with interest the manuscript # ijerph-841728 entitled “Age at diagnosis remains the main factor influencing the frequency and duration of partial remission in children newly diagnosed with type 1 diabetes: an observational study”

The aim of the present study was to define the prevalence and duration of partial remission in a cohort of paediatric patients suffering from type 1 diabetes. Secondary outcomes were to establish the influence of clinical and/or laboratory parameters on the occurrence and  length of the remission phase.

The authors showed that  63.5% of patients experienced partial remission. The mean duration of the remission phase in this study population was 13.4 months. Age was the main factor influencing both the occurrence and the duration of partial remission. These findings are very interesting and important to collect more information about this crucial phase in the clinical history of type 1 diabetes. I have only minor comments and points for discussion:

    The authors cited the criterion for defining the severity of diabetic ketoacidosis. However, data on HCO3 levels at diagnosis of type 1 diabetes were not reported and considered in study procedures.

Reply: We did not report data on HCO3 since they were incomplete, especially for patients who were initially treated for DKA in other hospitals.

    The authors mentioned the glucagon stimulation test for determining stimulated c-peptide levels. However, other methods for assessing the stimulated c-peptide levels exist. In addition, both baseline and stimulated C-peptide normal values should be added.

Reply: We are conscious that stimulated c-peptide can also be evaluated by other tests. However, the glucagon stimulation test is the most approved method in the pediatric population. The normal range of basal c-peptide levels and the interpretation of the glucagon stimulation test were added in the text, as suggested (lines 117-118).

    Did the authors assess c-peptide levels during the follow-up period?

Reply: Unfortunately we did not evaluate c-peptide levels during the follow-up period. This aspect could be an interesting suggestion for further studies.

    Several studies concerning partial remission have been published in the literature, but researchers often disagreed with the definition of partial remission. The different approaches used by authors could lead to different conclusions. Have the authors considered this aspect in the discussion of the manuscript?

Reply: We are agree with the reviewer. As mentioned in the new paragraph ‘definition of partial clinical remission’ we considered the IDAA1c definition since this measure represents the most validated and scientifically accepted. In fact, the most recent studies on the honeymoon considered this parameter to evaluate the presence of the remission phase (lines 134-139).

    Recently, it has been shown that the length of partial clinical remission phase is related to the association of autoimmune diseases with type 1 diabetes (doi: 10.1155/2020/2630827). This aspect should be discussed.

Reply: We are grateful for the suggestion. We mentioned it in the discussion section and added the reference as suggested (lines 283-285; ref #41).

    Please be consistent with references format (verify Journal rules for writing or not day and month of publication?; last page format : -0; -00; -000, ...?)

Reply: Amended. References format was properly edited.

Reviewer 2 Report

This is an interesting study that looks at the influence of the age of diagnosis for type 1 diabetes and partial clinical remission in children.  It is an important topic and as the authors note can influence clinical care.  However, there are a number of questions related to the methods that should be addressed to put the results into context as detailed below.  The characteristics of patients who have partial clinical remission may be a sign of a slower progressing disease.  This may also be associated with diagnosis of type 1 diabetes in adulthood. While an individual without DKA at diagnosis may be more likely to have PCR, there is no evidence presented in this study that these children were diagnosed earlier.

Title:  I recommend the authors revise and shorten the title.  Possible suggestion- "Influence of age on partial clinical remission among children with newly diagnosed type 1 diabetes".

Abstract: Define partial clinical remission.  Add the direction of the association for BMI, pH, c-peptide levels.  From the results presented, there is not evidence that the study supports diagnosing type 1 diabetes as early as possible.

Methods:  Please define where the study population is from.  How were patients selected for the study?  what was the response rate?  How do patients end up at the pediatric diabetes clinic?  is it only the more serious cases?  Was there loss to follow-up? How generalizable are these patients to other patients with type 1 diabetes?

Were all patients hospitalized?  How might this influence the results of the study?  (Since not every individual with type 1 diabetes is hospitalized at diagnosis, this may limit the generalizability of the results)

Include a paragraph in the methods clearly defining PCR.  Also, the terms "remitter" and "PCR" are used interchangabily throughout the paper.  Please select one and be consistent.

Were all patients enrolled in the study at time of diagnosis?  

The mean age of patients is 13 years and the mean age at diagnosis is 8 years.  Were patients enrolled 5 years after diagnosis?

How soon after diagnosis was PCR?  Did any patients experience PCR more than once during follow-up?

Table 2-  include the median and IQR for the continous values reported in the table. 

Was the c-peptide reported in table 2 at baseline?  how long after diagnosis?

Is there additional information on insulin dosage and type?  any patients with CGM?

Please include units for all measures.

For table 3- did you center the results?  For the continous variables (ie age) please include  the unit of the association.  Is it for one year of age?  

Table 4.  suggest interpreting the beta coeffiecent (ie for age a 2 year increase is associated with a XX increase in duration of PCR).

Discussion- much of the discussion focuses on these results supporting early diagnosis.  but that is not clearly detailed in the paper.  If that is one of the hypotheses, then I suggest the authors use the characteristics they hypothesize are related to early diagnosis, stratify the patients by these characteristics and then systematically look to see if there are differences in PCR.  

Table 2.  Include an overall estimate of prevalence of PCR.

Author Response

Reviewer # 2

This is an interesting study that looks at the influence of the age of diagnosis for type 1 diabetes and partial clinical remission in children.  It is an important topic and as the authors note can influence clinical care.  However, there are a number of questions related to the methods that should be addressed to put the results into context as detailed below.  The characteristics of patients who have partial clinical remission may be a sign of a slower progressing disease.  This may also be associated with diagnosis of type 1 diabetes in adulthood. While an individual without DKA at diagnosis may be more likely to have PCR, there is no evidence presented in this study that these children were diagnosed earlier.

Title:  I recommend the authors revise and shorten the title.  Possible suggestion- "Influence of age on partial clinical remission among children with newly diagnosed type 1 diabetes".

Reply: We are grateful for the suggestion. We changed the title as proposed (lines 1-2).

Abstract: Define partial clinical remission.  Add the direction of the association for BMI, pH, c-peptide levels.  From the results presented, there is not evidence that the study supports diagnosing type 1 diabetes as early as possible.

Reply: The definition of partial clinical remission was added (lines 14-16), as well as the direction of the association for BMI, pH and c-peptide levels (lines 24-25). The conclusion on the early diagnosis was deleted (lines 27-29 – “strike through” function in Microsoft Word).

Methods:  Please define where the study population is from.  How were patients selected for the study?  what was the response rate?  How do patients end up at the pediatric diabetes clinic?  is it only the more serious cases?  Was there loss to follow-up? How generalizable are these patients to other patients with type 1 diabetes?

Reply: We specified the geographic origin of our study population (lines 75-77). As mentioned, all the patients newly diagnosed with type 1 diabetes were enrolled in the study but only children and adolescents who were continuously followed-up in our Diabetes Centre, for at least two years, were included in the cohort study. We think that our study population could be assimilated to the paediatric population with type 1 diabetes.

Were all patients hospitalized?  How might this influence the results of the study?  (Since not every individual with type 1 diabetes is hospitalized at diagnosis, this may limit the generalizability of the results)

Reply: According to the Italian Society of Pediatric Endocrinology and Diabetes all the young patients with new diagnosis of type 1 diabetes are hospitalized (lines 93-95). Therefore, there are no selection bias that could influence the results.

Include a paragraph in the methods clearly defining PCR.  Also, the terms "remitter" and "PCR" are used interchangabily throughout the paper.  Please select one and be consistent.

Reply: The paragraph was added, as suggested (lines 134-139; deleted lines 91-92 and 104-105). In addition, we decided to use the term “PRC” throughout the paper.

Were all patients enrolled in the study at time of diagnosis?

Reply: All the patients were enrolled in the study at the time of diagnosis. We specified it in the text (lines 79-80). 

The mean age of patients is 13 years and the mean age at diagnosis is 8 years.  Were patients enrolled 5 years after diagnosis?

Reply: We referred to the mean age of patients at the time of the publication. In order to avoid misunderstanding, we decided to delete it and we only considered the mean age at the diagnosis (line 161 and Table 1).

How soon after diagnosis was PCR?  Did any patients experience PCR more than once during follow-up?

Reply: As mentioned in the paragraph “Definition of partial clinical remission” we evaluated PCR starting from the first follow-up visit, three months after the diagnosis (lines 138-139). To the best of our knowledge, PCR can occur only once in the course of type 1 diabetes clinical history.

Table 2-  include the median and IQR for the continous values reported in the table.

Reply: The median and IQR were added both in table 1 and in table 2.

Was the c-peptide reported in table 2 at baseline?  how long after diagnosis?

Reply: Basal and stimulated c-peptide reported in table 2 were at the onset. Both these levels were assessed after the acidosis had resolved and subcutaneous insulin therapy had been initiated, as already cited in the text (lines 112-114).

Is there additional information on insulin dosage and type?  any patients with CGM?

Reply: The mean insulin dosage at the discharge after the onset was already present (table 1 – table 2). We specified that all the patients started with multiple daily insulin injection regimen according to the “basal-bolus” scheme (lines 96-97). No patient started insulin therapy with CGM.

Please include units for all measures.

Reply: Units for all measures were included.

For table 3- did you center the results?  For the continous variables (ie age) please include  the unit of the association.  Is it for one year of age? 

Reply: In the table 3, in the univariate and multivariate models units of the association are expressed as odd ratios. All the variables of the models are considered in the same unit of measure, with which the data were collected (i.e. age of onset in years, HbA1C in % or mmol/mol, the total daily insulin in IU/Kg) but in this statistical test the evaluation of association is not related to the unit of measure.

Table 4.  suggest interpreting the beta coeffiecent (ie for age a 2 year increase is associated with a XX increase in duration of PCR).

In a regression model, the coefficients can be interpreted on the basis of their positive or negative sign. A positive coefficient indicates that as the value of the independent variable increases (i.e. age of onset, equal to 0.84), the mean of the dependent variable (PCR duration) also tends to increase: in particular, a unit increase in x results in an increase in average y by 0.84 units, all other variables held constant A negative coefficient suggests that as the independent variable increases (i.e. Total daily insulin, equal to -5.38), the dependent variable tends to decrease: a unit increase in x results in a decrease in average y by 5.38 units. In presence of a categorical variable, such as gender, we converted it into a dummy variable which takes values 1 for males and 0 for females: for the interpretation, we can say that average y (PCR duration) is higher by 1.21 units for males than for females, all other variables held constant.

Discussion- much of the discussion focuses on these results supporting early diagnosis but that is not clearly detailed in the paper.  If that is one of the hypotheses, then I suggest the authors use the characteristics they hypothesize are related to early diagnosis, stratify the patients by these characteristics and then systematically look to see if there are differences in PCR. 

Reply: Amended. We deleted the part of discussion on the hypothesis of early diagnosis (deleted lines 234-240 and 309-315 - “strike through” function in Microsoft Word).

Table 2.  Include an overall estimate of prevalence of PCR.

Reply: In according to your helpful suggestion, the number and percentage of patients who achieved or not the PCR was added in the first row of Table 2.
